# Anatomical Variants Identified on Computed Tomography of Oropharyngeal Carcinoma Patients

**DOI:** 10.3390/medicina59040707

**Published:** 2023-04-04

**Authors:** Sirorat Janta, Athikhun Suwannakhan, Laphatrada Yurasakpong, Arada Chaiyamoon, Nutmethee Kruepunga, Joe Iwanaga, R. Shane Tubbs, Pinthusorn Eiamratchanee, Tawanrat Paensukyen

**Affiliations:** 1Anatomy Unit, Department of Medical Science, Faculty of Science, Rangsit University, Pathum Thani 12000, Thailand; sirorat.j@rsu.ac.th; 2Department of Anatomy, Faculty of Science, Mahidol University, Bangkok 10400, Thailand; 3In Silico and Clinical Anatomy Research Group (iSCAN), Bangkok 10400, Thailand; 4Princess Srisavangavadhana College of Medicine, Chulabhorn Royal Academy, Bangkok 10210, Thailand; 5Department of Anatomy, Faculty of Medicine, Khon Kaen University, Khon Kaen 40002, Thailand; 6Department of Neurosurgery, Tulane University School of Medicine, New Orleans, LA 70112, USA; 7Department of Neurology, Tulane University School of Medicine, New Orleans, LA 70112, USA; 8Department of Anatomy, Kurume University School of Medicine, Fukuoka 830-0011, Japan; 9Department of Structural and Cellular Biology, Tulane University School of Medicine, New Orleans, LA 70112, USA; 10Department of Neurosurgery, Ochsner Neuroscience Institute, Ochsner Health System, New Orleans, LA 70112, USA; 11Department of Anatomical Sciences, St. George’s University, St. George’s FZ818, Grenada; 12St. George’s International School of Medicine Keith B. Taylor Global Scholars Program, Northumbria University, Newcastle-upon-Tyne NE7 7XA, UK; 13Biomedical Science Program, Faculty of Science, Mahidol University, Bangkok 10400, Thailand

**Keywords:** computed tomography, anatomical variations, head and neck, chest, oropharyngeal cancer

## Abstract

*Background and Objectives:* Anatomical variations in the head, neck and chest are common, and are observed as occasional findings on computed tomography (CT). Although anatomical variations are mostly asymptomatic and do not cause any negative influence on the body function, they may jeopardize diagnosis and may be confused with pathological conditions. The presence of variations may also limit surgical access during tumor removal. The aim of this study was to investigate the prevalence of six anatomical variations—os acromiale, episternal ossicles, cervical rib, Stafne bone cavity, azygos lobe and tracheal bronchus—in an open-access computed tomography dataset obtained from oropharyngeal cancer patients. *Materials and Methods:* A total of 606 upper-chest and neck computed-tomography scans (79.4% male and 20.6% female) were retrospectively investigated. Sex difference was evaluated using the z-test for two proportions. *Results:* Os acromiale, episternal ossicles, cervical rib, Stafne bone cavity, azygos lobe, and tracheal bronchus were present in 3.1%, 2.2%, 0.2%, 0%, 0.3% and 0.5%, respectively, of all patients. Os acromiale was identified as meso-acromion in 86.6%, and as pre-acromion in 17.4%, of all acromia. Episternal ossicles were present unilaterally in 58.3%, and bilaterally in 41.7%, of all sterna. Only the cervical rib showed a sex difference in prevalence. *Conclusions:* awareness of these variations is important for radiologists interpreting head, neck and chest CTs; for example, those of oropharyngeal cancer patients. This study also illustrates the applicability of publicly available datasets in prevalence-based anatomical research. While most of the variations investigated in the present study are well-known, the episternal ossicles are not well explored, and need further investigation.

## 1. Introduction

Anatomical variations observed in head, neck and chest CT are relatively common. Owing to their asymptomatic and sporadic presentation, these variants are typically incidental findings on CT and can be overlooked if not actively investigated. Awareness of them is vital and they should not be confused with pathological conditions. It has been reported than 10% of malpractice cases are attributable to ignorance of anatomical variations [1]. Under normal circumstances, the variations are asymptomatic and exert no negative influence on organ function. However, they can jeopardize diagnosis and medical procedures potentially requiring more attention or totally different arrangements [2], for example during surgical removal of the tumors. Therefore, identifying them affects clinical outcomes positively by minimizing potential confusion or malpractice.

Oropharyngeal cancer is the sixth most common malignancy worldwide [3]. Around 6% to 20% of patients diagnosed with this type of cancer have multiple primary tumors or distant metastases on initial presentation, often arising within the head and neck itself, though metastasis in the lungs or esophagus is possible [4]. It is crucial to identify all sites of primary tumors and distant metastases in newly diagnosed oropharyngeal cancer cases because they can have negative effects on management, prognosis and patient survival. Computed tomography (CT) is currently the gold standard for patients with suspected thoracic malignancy [4]. For most oropharyngeal cancer patients, chest malignancy is most common in the lung apices or mediastinal lymph nodes. As a result, a limited chest CT from the thoracic inlet to the pulmonary veins level is performed. Recently, developments in artificial intelligence have led to phenomenal growth in the automated diagnosis of cancers [5]. This requires an extensive repertoire of images to improve performance and accuracy, so an increasing number of cancer-related CT datasets have been made publicly available by uploading them into online depositories such as The Cancer Imaging Archive (TCIA). In addition to their intended uses, Yurasakpong et al. [6] recently proposed that these datasets could be a valuable resource for anatomical research, and they can be used to examine potential relationships between variant anatomy and predisposition to diseases.

Several anatomical variations can occur in the head, neck and upper thorax, including os acromiale (OA), episternal ossicles (EOs), cervical rib (CR), Stafne bone cavity (SBC), azygos lobe (AL), and tracheal bronchus (TB). The OA is a condition where the acromion bone of the shoulder blade fails to fuse during development, resulting in a separate bone fragment in around 7% of the general population [7]. The EOs are less well-known small accessory bones located at the posterior side of the manubrium present in 1.2% to 6.9% of individuals [8,9,10,11]. The CR, found in 1.1% of individuals [12], is an extra rib from the seventh cervical vertebra which is strongly associated with the development of thoracic outlet syndrome. The SBC is an extremely rare bone defect that can occur in the lingual surface of the mandible found in 0.17% of the population [13]. The AL is a rare (0.3% of the population) accessory lobe of the right lung, in which the upper lobe is divided by a misplaced azygos vein [14]. With a prevalence of 1.0% [15], the TB is a rare anatomical variant where an additional bronchus arises from the trachea before it enters the lungs. While many of these variations are asymptomatic and do not require treatment, they can sometimes cause discomfort, compression of nearby structures, or confusion in the interpretation of imaging studies.

Therefore, the aim of this study was to investigate the prevalence of six well-known anatomical variants in the upper thorax, head and neck, using CT images obtained from an open-access oropharyngeal carcinoma computed tomography dataset.

## 2. Materials and Methods

The study was performed according to the Declaration of Helsinki, and was exempted by the Research Ethics Office of Rangsit University (RSU-ERB2023-007) on 3 March 2023.

### 2.1. Imaging Dataset

The CT images used in this study were obtained from the Radiomic Biomarkers in the Oropharyngeal Carcinoma (OPC-Radiomics) dataset [16], which was available on the TCIA website. This dataset contains unenhanced head and neck CTs from 606 (79.4% male and 20.6% female) human papillomavirus-related oropharyngeal cancer patients. The average age was 60.5 ± 9.9 years (range 33–89 years). All images were in DICOM format with 512 × 512-pixel resolution. The sex and age of each patient were known, but other patient-specific information was blinded.

### 2.2. Image Analysis

Six anatomical variations were examined including the OA, EOs, CR, SBC, AL and TB. The OA was identified on axial views as an unfused segment of the acromion and was classified into three types: pre-acromion, meso-acromion and meta-acromion. The EOs were seen as small ossicles posterior to the manubrium on the axial view and were classified as left, right or bilateral. The CR was observed on axial views as an additional rib originating from the seventh cervical vertebra. The mandible was inspected on axial views for an SBC, or a well corticated as a concavity on the lingual surface of the mandible. The right lung was examined for an AL by identifying the azygos fissure on the axial view. A TB was observed on the coronal view and identified as a supernumerary branching of the trachea above the carina. All variants were identified by two authors with PhDs in anatomy. Any disagreement was resolved by discussion and consultation with an expert radiologist. Three-dimensional (3D) images of these variants were reconstructed on 3D Slicer using segment editor [17]. Sex differences were analyzed using a *z*-test for two proportions. Cohen’s ϰ was used to evaluate the level of agreement between the two observers. All statistical analyses were performed using Stata 17 (StataCorp, College Station, TX, USA). Statistical significance was accepted at *p* = 0.05 (two-tailed).

## 3. Results

### 3.1. Data Exploration and Exclusion

There were 606 CT scans in the dataset. Sixteen patients were excluded from the analysis of EOs because they had undergone median sternotomy. Two were excluded from the prevalence analysis of TB because no tracheal bifurcation was visible on the CT. The sternalis muscle, cardiac bronchus, sternal foramen and xiphoid foramen were not investigated because only the upper third of the chest was visible on the CT.

### 3.2. Prevalences of Six Anatomical Variants

The prevalence of the six anatomical variations are presented in Table 1. Cohen’s ϰ value was 0.94 for all variants. However, after discussion and consultation with an expert radiologist, Cohen’s ϰ reached 1.00. The OA was observed in 20 patients, a crude prevalence of 3.3%. It was bilateral in 5 patients (25%) and unilateral in 15 (75%). Out of 1212 clavicles investigated, the true prevalence of OA was 1.9% (23 cases). The OA was classified as meso-acromion (Figure 1A) and pre-acromion (Figure 1B) in 19 (82.6%) and 4 (17.4%) acromia, respectively. The meta-acromion type was not observed.

The EOs were present at the posterior surface of the manubrium in 12 cases (2.0%). It was unilateral in seven sterna (58.3%) (Figure 2A,B) and bilateral in five (41.7%) (Figure 2C). The CR was found bilaterally in one patient (0.2%) (Figure 3). In this patient, the left CR was approximately 40 mm long and did not articulate with any other structure. The right CR was approximately 60 mm long. It coursed anteriorly and inferiorly to reach the manubrium, but did not articulate with it.

The TB and AL were observed in three (0.5%) and two (0.3%) patients, respectively. In all TB cases, the TB supplied the apical segment of the superior lobe of the right lung. No SBC was observed in any patient (Figure 4A,B). In one patient with an AL, the superior lobe of the right lung was unusually small and the anterior margin of the oblique fissure was located aberrantly superior (Figure 4C,D). There was a statistically significant sex difference in CR prevalence (*p* = 0.05). No significant sex differences were found for other variants (Table 1).

## 4. Discussion

Traditionally, radiological studies are performed using institutionally provided images and require prior ethical approval. Despite the challenges caused by the COVID-19 pandemic, there is a window of opportunities for data-driven research, particularly in the field of artificial intelligence and automated diagnosis. It was recently proposed by Yurasakpong et al. [6] that these datasets could also be used for anatomical research to study disease-related alterations of normal anatomy.

Head and neck cancers are associated with high mortality, because they develop without obvious symptoms, leading to late diagnosis in advanced stages. Artificial intelligence technology has made significant progress in the automated diagnosis of medical images, which could detect cancer early [18]. In addition, this technology could also have significant implications in predicting treatment response, disease progression, and patient survival in clinical settings [19]. As a result, more and more imaging datasets have been uploaded to online repositories to fuel research related to automated diagnosis and artificial intelligence [20]. For example, TCIA is a freely accessible resource that contains a vast collection of medical imaging data sets used for research, education, and development of cancer treatments [21]. The TCIA currently offers 186 imaging data sets from various modalities, including CT, magnetic resonance imaging, and ultrasound.

The presence of skeletal variations may alter the process of automatic diagnosis using machine learning, because they may be confused with metastatic bone change. Even though bone metastasis is considered rare in head and neck cancers and only appears in the late stages, its incidence was 50% or as high as 80% in nasopharyngeal carcinoma patients [22,23]. Tumor metastasis to bone caused a disturbance in bone remodeling and imbalances in the osteoclast and osteoblast [24]. Osteolytic lesions are characterized by the destruction of bone tissue, which leads to bone resorption and loss of bone density. On the other hand, osteoblastic lesions are characterized by an increase in bone density due to the formation of new bone tissue [24]. Accessory bones such as the EOs may mimic osteoblastic lesion. In addition, osteolytic lesion in the mandible has been reported previously [25], with an appearance similar to that of the SBC when seen on panoramic radiograph.

In the present study, the OPC-Radiomics dataset was explored and six anatomical variations in the head, neck and upper thorax, were investigated. Our findings are also presented three-dimensionally, which we believe will be standard in future anatomical studies [26]. Four skeletal variants were examined: the OA, EOs, CR and SBC. We found the OA in 3.1% of subjects. The meso-acromion type was almost five times more common than the pre-acromion type. A meta-analysis revealed an overall OA prevalence of 7.0%, but its prevalence in radiological studies was only 4.2% [7]. Similar to previous studies, the meso-acromion type accounted for 76.6% of the total OA cases. Although neither sex nor side affected its incidence, the OA was significantly more prevalent in African populations [7]. Two theories were proposed to explain the occurrence of the OA. It is believed that the OA is formed when any of the three ossification centers of the acromion fail to fuse with the basi-acromion. These ossification centers begin to develop around 14 to 16 years of age, and are completely fused at 18 to 25 years [27]. In the present study, the possibility of incomplete fusion was ruled out, because the minimum age of the subjects was 33 years. Angel et al. [28] suggested a genetic explanation for the high rate of OA found in 30% of the remains from the cemetery of the First African Baptist Church in Philadelphia. The burial groups and spatial proximity of the skeletons exhibiting OA in the cemetery could indicate familial and possibly genetic links [28]. The EOs, sometimes known as suprasternal bones, are less well-known anatomical variants and have been investigated in relatively few studies; their prevalence ranges from 1.2% to 6.9% [8,9,10,11]. In our study, their prevalence was 2.2%. They were predominantly found on the left side when they were present unilaterally. The EOs should not be confused with fractures, calcified lymph nodes or vascular calcifications. It is believed that these ossicles develop as rudiments of the epicoracoids of the primitive shoulder because of their constant attachment to the interarticular disk of the sternoclavicular joint [11]. The CR was observed bilaterally (0.2%) in one patient in the present study, which is lower than the global prevalence of 1.1%, according to a meta-analysis [12]. This variant indicates persistent ossification of the lateral costal element of the C7 vertebra. The CR, a supernumerary rib above the first rib, is 25 times more common in thoracic outlet syndrome patients, owing to potential compression of the neurovascular bundle [12]. Management options for thoracic outlet syndrome caused by a CR include anterior scalenectomies and rib resections [29]. The SBC or lingual mandibular bone defect is a rare variation, and was not observed in this study, possibly because the sample was too small. A recent meta-analysis reported the SBC in 0.12%; it should be regarded as a variation rather than a defect, because of its asymptomatic nature [13]. Although the occurrence of SBC has been extensively studied, its cause is still not certain [30]. Some authors believe it is present from birth, while others think it develops later in life. Those who support the former view suggest that SBC arises from a portion of the submandibular gland being trapped during the development of the jawbone, resulting in a distinct cavity with ectopic salivary glands [31,32]. However, the fact that SBC is more common in older adults than in children [33] challenges this idea, suggesting that it may develop later in life. Some authors suggest that local pressure from the submandibular or sublingual gland’s growth may cause SBC [34], but this theory is also unproven.

Two bronchopulmonary variants were studied: the AL and TB. There was an AL, or accessory lobe of the superior lobe of the right lung, in 0.3% of cases in this study. This is identical to the average prevalence of 0.3% reported in a recent meta-analysis [14]. In one of our two cases, the AL was accompanied by a superiorly placed oblique fissure, which downsized the superior lobe of the right lung (Figure 4C,D). This variant oblique fissure should not be confused with an azygos fissure, because no azygos vein was found in it. The AL is nine times more common in people with congenital abnormalities, suggesting that its presence is genetically mediated [14]. It is currently accepted that the AL is formed when the azygos vein precursor migrates to its usual position through, rather than medial to, the apex of the right lung [35]. The appearance of an AL on a CT can be mistaken for an abscess, bulla, or pneumothorax [36]. The TB, found in 0.5% of cases in this study, represents an accessory bronchus that originates superior to the tracheal bifurcation. It usually gives rise to the apical bronchopulmonary segment of the superior lobe of the right lung. Three hypotheses have been proposed to explain its formation: reduction of a previously developed bronchus, migration of part of the developed hyparterial branching pattern to a different location, either on the trachea or a bronchus, and local morphogenesis [37]. A meta-analysis reported that the overall prevalence of TB was 1.0% [15]. Similar to the AL, the TB was 15 times more common (14.9%) in individuals with congenital anomalies, which suggests that TB occurrence may be influenced by genetics.

This study has a number of limitations. Although most skeletal variations were investigated, atlas arch defects were intentionally not included, because they are part of another unpublished study. While the use of open-access resources in anatomical research can lead to numerous research opportunities, sampling bias is inevitably introduced because these scans were obtained from oropharyngeal cancer patients and are not necessarily generalizable to the normal population. Subject demography is heavily biased towards males, because 90% of oropharyngeal cancers occur in males [38].

## 5. Conclusions

The prevalence of six well-known anatomical variants were investigated in this study. The prevalences were in the same range as those reported by other previous studies. Having certain anatomical variants does not modify the risk for oropharyngeal carcinoma, or vice versa. Awareness of these variations is essential for radiologists interpreting head, neck and chest CTs. A lack of understanding of them could lead to misdiagnosis and improper courses of treatment. Although most of the anatomical variants investigated in this study are well-known, the prevalence of episternal ossicles has only been studied by a small number of studies, and requires further investigation. In addition, this study illustrates the applicability of open-access datasets to anatomical research.

## Figures and Tables

**Figure 1 medicina-59-00707-f001:**
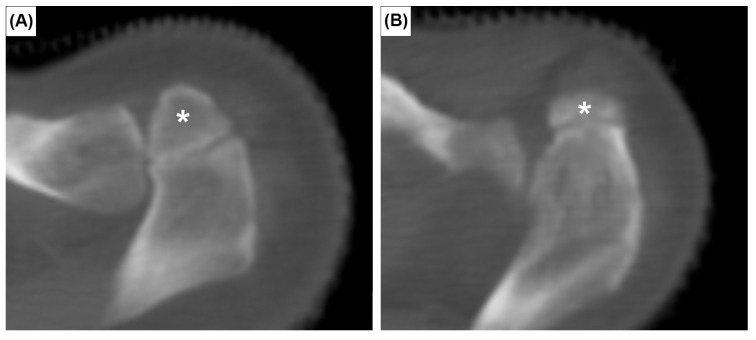
Axial CT images showing two types of os acromiale: meso-acromion (**A**) and pre-acromion (**B**). Asterisks indicate the os acromiale.

**Figure 2 medicina-59-00707-f002:**
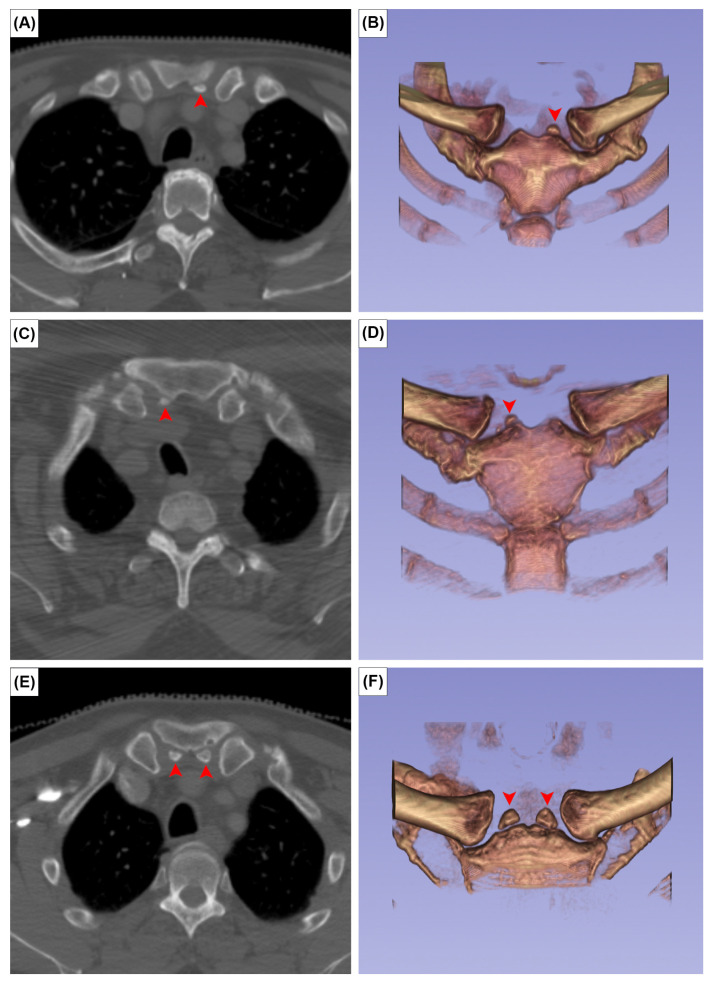
Axial CT images (**A**,**C**,**E**) and 3D reconstruction in anterior view of (**B**,**D**,**F**) the episternal ossicles, left-sided (**A**,**B**), right-sided (**C**,**D**), and bilateral (**E**,**F**) types. Red arrowheads indicate the episternal ossicles.

**Figure 3 medicina-59-00707-f003:**
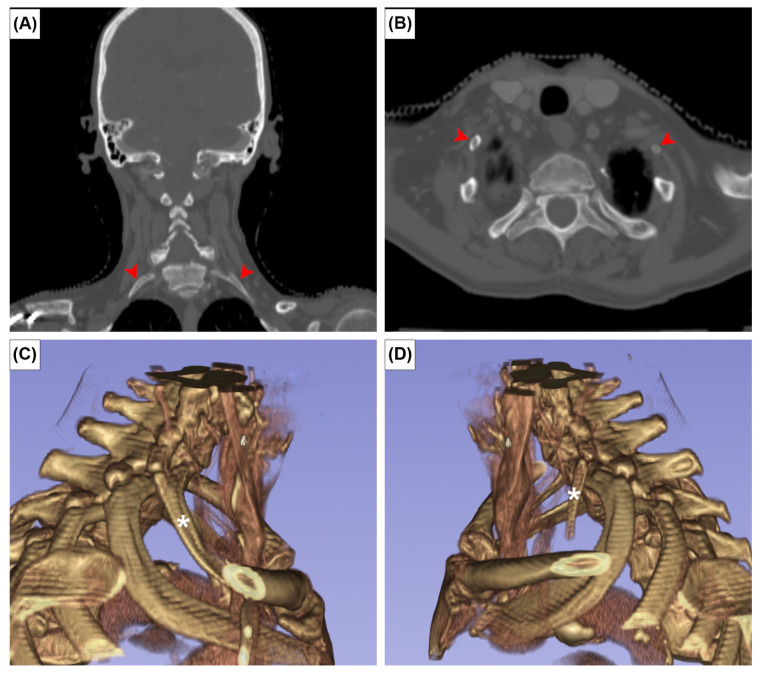
Coronal (**A**), axial CT images (**B**) and three-dimensional reconstruction of right (**C**) and left (**D**) cervical ribs. Red arrowheads and asterisks indicate the cervical ribs.

**Figure 4 medicina-59-00707-f004:**
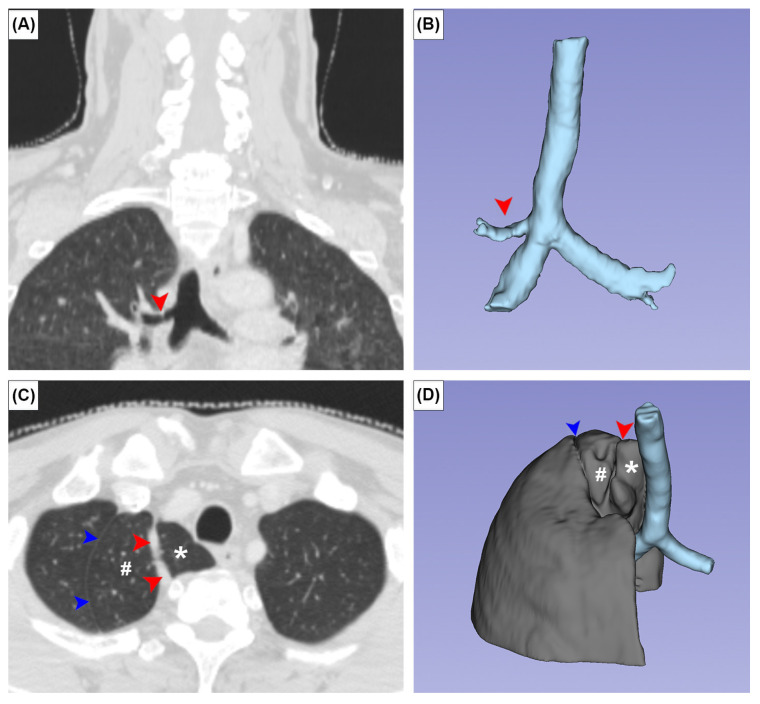
Coronal (**A**,**C**) CT images and three-dimensional images of the tracheal bronchus (**A**,**B**) and azygos lobe (**C**,**D**). Red arrowheads and asterisks indicate the azygos fissure and azygos vein, respectively. Blue arrowheads and # indicate a variant placement of the oblique fissure and unusually small superior lobe of the right lung.

**Table 1 medicina-59-00707-t001:** Overall prevalence of anatomical variations observed in OPC-Radiomics computed-tomography dataset. *p*-values indicate sex differences. All prevalences are crude unless stated otherwise.

Structure and Types	Overall Prevalence	Sex-Based Prevalence (%)		*p*-Values
		Male	Female	
Os Acromiale (OA)	19/606 (3.1%)	18/481 (3.7%)	1/125 (0.8%)	0.09
Os acromiale (OA) (true)	23/1212 (1.9%)	N/A	N/A	-
Left	9/19 (47.4%)	9/18 (50%)	0/1 (0%)	-
Right	6/19 (31.6%)	6/18 (33.3%)	0/1 (0%)	-
Bilateral Pre-acromion (true)	4/19 (21.0%)	3/18 (16.7%)	1/1 (100%)	-
	4/23 (17.4%)	N/A	N/A	-
Meso-acromion (true)	19/23 (82.6%)	N/A	N/A	-
Meta-acromion (true)	0/23 (0.0%)	N/A	N/A	-
Episternal ossicles (EOs)	14/590 (2.2%)	12/466 (2.6%)	2/124 (1.6%)	0.49
Left	6/14 (42.9%)	5/11 (45.5%)	1/3 (33.3%)	-
Right	3/14 (21.4%)	2/11 (18.2%)	1/3 (33.3%)	-
Bilateral	5/14 (35.7%)	4/11 (36.4%)	1/3 (33.3%)	-
Cervical rib (CR)	1/606 (0.2%)	0/481 (0%)	1/125 (0.8%)	0.05 *
Stafne bone cavity (SBC)	0/606 (0.0%)	0/481 (0%)	0/125 (0%)	-
Azygos lobe (AL)	2/606 (0.3%)	1/481 (0.2%)	1/125 (0.8%)	0.30
Tracheal bronchus (TB)	3/604 (0.5%)	2/480 (0.4%)	1/124 (0.8%)	0.58

* (asterisk) indicates significant difference between males and females; N/A, not applicable.

## Data Availability

The data that support the findings of this study are available from the corresponding author upon reasonable request.

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
