# Peer review of "Anatomical Variants Identified on Computed Tomography of Oropharyngeal Carcinoma Patients"

_medicina, 2023, doi:10.3390/medicina59040707_

Round 1

Reviewer 1 Report

Abstract- Material and methods - which are statistical tests used for the analysis

Abstract -line 36- I think it is about prevalence and not incidence of the cervical rib

Introduction - to specify which are the 6 types of anatomical variants referred to

Include future recommendations of the study

Author Response

Dear reviewer,

Thank you for your careful evaluation of our work. The manuscript has been modified in light of the reviewer’s suggestion. Please find below our point-by-point replies. 

Best wishes,

The authors

 Abstract- Material and methods - which are statistical tests used for the analysis

Authors’ reply: We have added “z-test for two proportions” to the abstract. Thank you.

Abstract -line 36- I think it is about prevalence and not incidence of the cervical rib

Authors’ reply: Thank you. We have changed the term “incidence” to “prevalence” as requested.

Introduction - to specify which are the 6 types of anatomical variants referred to

Authors’ reply: We have added a paragraph to briefly explain all six variants.  

Include future recommendations of the study

Authors’ reply: Future recommendation has been added to the concluding sentences both in the abstract and conclusion section.

Reviewer 2 Report

General comments 

The purpose of the article is to describe the prevalence of anatomical changes found in patients with oropharyngeal carcinoma. This work does not represent novelty in the scientific field and does not go into great detail about how knowing these anatomical variations would help in the management of patients with oropharyngeal carcinoma. The major strength is the large sample size examined. However, the critical issues are enormous, and major revision would be appropriate.

Specific comments

- The abstract (maybe, the whole article) should be completely rewritten; it would be more interesting if the authors, after describing which anatomical variants are best known in the literature, evaluate or discuss what the advantages of such knowledge may be in the management of oropharyngeal carcinoma. As a neuroradiologist, I find difficult to imagine a supernumerary bone or venous variant as a disturbing factor in a diagnosis of oropharyngeal carcinoma. Instead, I think it is more interesting to consider how the presence of such variants may alter surgical access to such tumors.

- In introduction section, the advent of artificial intelligence for automatic diagnosis of head and neck cancers is mentioned. With that purpose? It would be useful to review the literature to look for whether anatomical variants can alter the process of machine learning. Regarding the role of machine learning in head and neck tumors, I suggest these doi articles which could be discussed in discussion section: 10.3389/fonc.2015.00272;  10.3390/cancers15041174; 10.1016/j.mric.2021.06.016.

- material and methods/results: why did you evaluate only six anatomical variants? by what criterion did you choose these variants? were other variants also evident? it would be good to know an ICC between the two observers; possibly, it would be interesting to know whether the radiologist is capable of recognizing a greater number of variants than operators with PhDs in anatomy in order to assess any false negatives.

Author Response

Dear reviewer,

Thank you for your careful evaluation of our work. The manuscript has been modified in light of the reviewer’s suggestion and we hope that the manuscript is now acceptable.

Please find below our point-by-point replies.

Best wishes,

The authors

General comments 

The purpose of the article is to describe the prevalence of anatomical changes found in patients with oropharyngeal carcinoma. This work does not represent novelty in the scientific field and does not go into great detail about how knowing these anatomical variations would help in the management of patients with oropharyngeal carcinoma. The major strength is the large sample size examined. However, the critical issues are enormous, and major revision would be appropriate.

Authors’ reply: We highly appreciate your comments which have improved the quality of our work significantly. All comments have been reflected in this revised version. We hope that the manuscript is now acceptable for publication.

Specific comments

- The abstract (maybe, the whole article) should be completely rewritten; it would be more interesting if the authors, after describing which anatomical variants are best known in the literature, evaluate or discuss what the advantages of such knowledge may be in the management of oropharyngeal carcinoma. As a neuroradiologist, I find difficult to imagine a supernumerary bone or venous variant as a disturbing factor in a diagnosis of oropharyngeal carcinoma. Instead, I think it is more interesting to consider how the presence of such variants may alter surgical access to such tumors.

Authors’ reply: Thank you for your valuable comments. We have rewritten the abstract to highlight that variations may alter surgical access to tumors. At the end of the abstract, we have indicated that among the variations that were explored, episternal ossicles are less well-known and need further investigation to better understand its prevalence and etiology.

- In introduction section, the advent of artificial intelligence for automatic diagnosis of head and neck cancers is mentioned. With that purpose? It would be useful to review the literature to look for whether anatomical variants can alter the process of machine learning. Regarding the role of machine learning in head and neck tumors, I suggest these doi articles which could be discussed in discussion section: 10.3389/fonc.2015.00272;  10.3390/cancers15041174; 10.1016/j.mric.2021.06.016.

Authors’ reply: Thank you for raising this. We have added a paragraph along with the suggested references to discuss the use of artificial intelligence and whether it could alter the process of automatic detection. One important information we would like to highlight is that extra bone mass may mimic osteoblastic lesion which might have an effect on machine learning performance. Likewise, osteolytic lesion of the mandible also appears quite similar to a Stafne bone defect when seen on CT or panoramic radiographs (https://doi.org/10.1016/j.bjoms.2017.08.219). These examples have been added. However, for coherence we have decided to add these points to the discussion instead of the introduction. Thank you for your understanding. 

- material and methods/results: why did you evaluate only six anatomical variants? by what criterion did you choose these variants? were other variants also evident? it would be good to know an ICC between the two observers; possibly, it would be interesting to know whether the radiologist is capable of recognizing a greater number of variants than operators with PhDs in anatomy in order to assess any false negatives.

Authors’ reply: Thank you for raising this important comment. These six structures were chosen because they are the only variants that are visible from the upper one-third of the chest and upward., The other variant that could be investigated is atlas arch defect. However, this is part of another anatomical study currently being considered by another journal. This issue has been added as a limitation. Other variants including sternalis muscle, cardiac bronchus, sternal foramen and xiphoid foramen were not investigated because only the upper one-third of the chest was visible on CT. Sternalis muscle was visible in some cases but not the whole muscle length, so we decided not to study it. We already provided such an explanation in the first paragraph of the results section. 

To assess agreement between the two observers, we have added Cohen’s kappa to the methods section to evaluate inter-rater reliability. There were only 11 disagreements out of 606 CT scans, which yielded a kappa value of 0.94. However, after discussion among the two anatomists with consultation from an expert radiologist, the kappa value reached 1.00. The agreement was very high because it was not the first time that we investigated these variations. Our group published a similar paper last year and the EOs, AL and TB were investigated (https://onlinelibrary.wiley.com/doi/abs/10.1002/ca.23873). The most problematic structure was os acromiale because in 3-4 cases the synchondrosis between os acromiale and the rest of the clavicle was not clearly visible, this is when the expert had to weigh in. Also, the expert helped us a lot with confirmation of the azygos lobe because the case that we encountered was associated with a variant oblique fissure (Figure 4C-D). It was confusing at first on axial CT because there are two fissures (Figure 4C) but we learned it was not that uncommon. Concerning the false negatives, we were less concerned about it because the two anatomists were too “sensitive” and had a tendency to yield more false positives, which were easily resolved.

Round 2

Reviewer 2 Report

The Authors have addressed the comments and the manuscript can now be accepted.